# Treatment Modalities for Angina with Non-Obstructive Coronary Arteries (ANOCA): A Systematic Review and Meta-Analysis

**DOI:** 10.3390/jcm14124069

**Published:** 2025-06-09

**Authors:** Fabienne E. Vervaat, Annemiek de Vos, Jimmy Schenk, Pim A. L. Tonino, Inge F. Wijnbergen

**Affiliations:** 1Department of Cardiology, Catharina Hospital, 5623 EJ Eindhoven, The Netherlands; 2Department of Methodology, University of Amsterdam, 1105 AZ Amsterdam, The Netherlands

**Keywords:** angina with non-obstructive coronary arteries (ANOCA), treatment modalities, meta-analysis

## Abstract

**Background and Objectives:** Up to 40% of patients undergoing a coronary angiogram due to angina pectoris have no obstructive coronary artery disease, also known as angina with non-obstructive coronary arteries (ANOCA). ANOCA is associated with significant impairment in patients’ quality of life, increased risk of myocardial infarction and all-cause mortality. Approximately 25% of patients with ANOCA have persisting symptoms despite optimal medical therapy. There is a lack of in-depth knowledge regarding tailored treatment for patients with ANOCA due to a scarcity of trials designed to assess the effect of treatment modalities. The aim of this systematic review and meta-analysis is to give clinicians an overview of the efficacy of current treatment modalities for patients with ANOCA. **Methods:** PudMed/MEDLINE, Embase, the Cochrane Library and clinical trial registries were searched for randomised controlled and cohort studies regarding treatment modalities for ANOCA. The main outcome was change in angina pectoris frequency for each treatment modality. Secondary outcomes included changes in exercise capacity, quality of life, Canadian Cardiovascular Society (CCS) class, coronary flow reserve (CFR) and survival. **Results:** In total, 80 studies were included and used in the meta-analysis, of which ten studies met the current definition of ANOCA. Angina pectoris frequency improved significantly in the majority of the treatment modalities, with neuromodulation resulting in −3.35 standardised mean difference (SMD) (95% CI: −5.13; −1.56), trimetazidine in −1.74 SMD (−2.63; −0.85), traditional Chinese medicine in −1.55 SMD (−2.36; −0.75), beta-blockers in −1.32 SMD (−1.88; −0.77), enhanced external counterpulsation in −1.27 SMD (−2.04; −0.49), stem cell therapy in −1.04 SMD (−1.51; −0.57), lifestyle interventions in −0.86 SMD (−1.15; −0.57), RAAS-inhibitors in −0.83 SMD (−1.31; −0.35) and calcium channel blockers in −0.64 SMD (−0.92; −0.35). **Conclusions:** This meta-analysis into treatment modalities for patients with ANOCA shows a significant improvement in angina pectoris frequency in the majority of included treatment modalities. However, these results should be interpreted cautiously, as only ten of the studies included in the meta-analysis meet the current definition of ANOCA. This review underlines the importance of undertaking new studies with existing treatment modalities to determine the efficacy in patients with ANOCA.

## 1. Introduction

In up to 40% of patients undergoing a coronary angiogram (CAG) due to angina pectoris, no obstructive coronary artery disease (CAD) is found [1], also known as angina with non-obstructive coronary arteries (ANOCA). ANOCA is caused by dysfunction of the coronary vasomotor function in the vast majority (60–90%) of cases [2]. ANOCA can be divided into two endotypes, which can co-occur: (i) microvascular angina (MVA), caused by a combination of structural microcirculatory remodelling and functional arteriolar dysregulation, and (ii) vasospastic angina (VSA), caused by epicardial coronary artery spasm occurring when a hyper-reactive epicardial coronary segment is exposed to a vasoconstrictor stimulus [3]. Both endotypes of ANOCA are associated with a significantly increased risk of myocardial infarction (MI) and all-cause mortality [4], have a significant impact on quality of life [5] and result in increased health care resource utilisation [6].

In accordance with the 2024 ESC guidelines on chronic coronary syndromes [7], CAG with coronary functional testing (CFT) has to be performed to confirm the ANOCA diagnosis. MVA (i) is diagnosed when fractional flow reserve (FFR) > 0.8, coronary flow reserve (CFR) < 2.0, index of microvascular resistance (IMR) ≥ 25, or the hyperaemic myocardial velocity resistance (HMR) ≥ 1.9, combined with a negative acetylcholine test. VSA (ii) is diagnosed when FFR > 0.8, CFR ≥ 2.0, IMR < 25, and HMR < 1.9, combined with a positive acetylcholine test. The acetylcholine test is positive if the coronary artery diameter decreases more than 90%, angina occurs, and ischaemic electrocardiograph (ECG) changes occur.

Currently, treatment modalities for patients with MVA and/or VSA are lacking. Standard of care includes lifestyle changes, risk factor management and medical therapy including beta-blockers, calcium channel blockers, long-acting nitrates and/or nicorandil [3,7,8]. However, when these treatment strategies are employed, approximately 25% of ANOCA patients experience refractory angina symptoms [3]. Furthermore, it is noted that there is a lack of in-depth knowledge regarding tailored treatment for patients with ANOCA [3]. A key contributing factor to this lack of knowledge has been the absence of a uniform definition for ANOCA until 2020, which has led to a scarcity of trials designed to look at treatment modalities for this specific patient population.

Prior to 2020, a single definition for angina with non-obstructive coronary artery disease was lacking, resulting in the usage of terms such as cardiac syndrome X (CSX). Studies have investigated the efficacy of various treatment modalities in patients with CSX, which is now known as ANOCA. The aim of this systematic review is to provide an overview of the efficacy of different treatment modalities for patients with ANOCA including studies performed prior to 2020 using the term CSX.

## 2. Methods

This systematic review was performed according to the methodologic guidelines outlined in the Cochrane handbook for systematic reviews [9] and in adherence to the 2020 PRISMA statement [10]. The study was registered on the international prospective register of systematic reviews PROSPERO with the unique identifier CRD42023451317 prior to the systematic search of the literature.

### 2.1. Eligibility Criteria

All studies on the subject ANOCA, or, if prior to 2020, ‘normal coronary arteries and angina’ (see below for definition, Table 1), and regarding treatment modalities for ANOCA (see below for definition) were reviewed and assessed by two independent reviewers (F.V. and I.W.) on title and abstract. With regard to the definition ‘normal coronary arteries and angina’, only studies with underlying microvascular angina were included; studies with a non-coronary cause of angina were excluded. Studies were included irrespective of publication status or language of publication. Studies examining the general adult human population or healthy adult humans (>18 years) were included. The study designs that were included were randomised controlled trials of any design and observational studies.

### 2.2. Outcome Measures

#### 2.2.1. Primary Outcome Measure

To determine the efficacy of treatment modalities in patients with ANOCA with regard to improvement in angina pectoris frequency (measured using scales and/or questionnaires such as the Seattle Angina Questionnaire (SAQ) and/or Numeric Rating Scale (NRS)) in comparison to placebo, comparator treatment modalities or baseline (prior to starting relevant treatment modality).

#### 2.2.2. Secondary Outcome Measures

To determine the efficacy of treatment modalities in patients with ANOCA compared to placebo, comparator treatment modalities or baseline, with regard to (i) exercise capacity, measured using bicycle ergometry and/or 6 min walking test, (ii) quality of life, measured using questionnaires such as the SAQ, RAND-36, 36-SF and/or EQ-5D, (iii) Canadian Cardiovascular Society (CCS) class, (iv) changes in CFR, coronary blood flow (CBF) and/or myocardial perfusion reserve (MPR), measured by CFT, PET, MRI or other imaging modality, and (v) survival, measured as a dichotomous value.

### 2.3. Search Strategies

#### 2.3.1. Database Search

PubMed (MEDLINE), Embase (Elsevier) and Cochrane library were searched from inception to 21 November 2024. One of the review authors (F.V.) designed the search strategy in PubMed and pilot tested with one other review author (I.W.), followed by translation for the other databases. The complete search strategy for all databases is provided in the Appendix A. Literature saturation was ensured by identifying relevant studies in the reference lists of included studies, and the clinical trial registers (ClinicalTrials.gov and ICTRP portal) were searched to identify additional studies. Conference abstracts were excluded unless there was another public report with additional information, and/or the authors were contacted for additional information.

#### 2.3.2. Study Selection and Screening

Due to the expected large number of hits based on the broad search strategy, one reviewer (F.V.) screened the title and abstract of the retrieved records against the inclusion criteria. The screening was performed using the internet-based software program Rayyan.qcri. Full reports of the titles that appeared to meet the inclusion criteria were obtained, and these full text reports were screened independently by two reviewers (F.V. and I.W.). Disagreements were identified using Rayyan.qcri and resolved by discussion or by consulting a third author (A.V.).

### 2.4. Data Extraction

Two independent reviewers (F.V. and I.W.) extracted the following data using a data extraction form (built in Excel):-Study characteristics: author, publication year, study design, enrolment period and follow-up duration.-Participants’ characteristics: total study population, age of the study population, gender, method used to diagnose ANOCA or, if prior to 2020, ‘normal coronary arteries and angina’ and endotype.-Intervention description: type of treatment modality (decision set) and comparator treatment modality (including placebo or baseline; supplementary set).

Outcome measures: interventional effect and standard error (SE).

### 2.5. Assessment of Risk of Bias in Included Studies

Two reviewers (F.V. and I.W.) independently assessed the risk of bias for each included study. The randomised controlled trials were assessed using the ‘Risk of Bias tool’ described in the *Cochrane Handbook for Systematic Reviews of Interventions* for randomised studies, RoB 2 [11]. For the non-randomised studies, the Newcastle–Ottawa Scale (NOS) was used to assess risk of bias [12]. If disagreements arose, they were resolved by discussion or by consulting a third author (A.V.).

### 2.6. Measurement of Effect and Data Synthesis

For the outcome data, continuous variables were analysed using weighted mean differences with 95% CI or standardised mean differences with 95% CI if different measurement scales were used. The calculated SMD was transformed back into the scale that was most frequently used in the included studies to allow clinical interpretation of the effect. Dichotomous variables were determined by using the risk ratio (RR) with 95% confidence interval (CI) if the data were prospectively collected or the odds ratio (OR) with 95% CI if the data were retrospectively collected.

Meta-analysis was only undertaken if two or more studies reported the same outcome whilst applying the same treatment modality. A random effects model (DerSimonian and Laird random effects method) was applied if the assumption of heterogeneity was met. Heterogeneity was defined as at least two of the following: (i) statistically significant Q-test (*p*-value < 0.1), (ii) an I^2^ statistic of >40% and (iii) prediction interval including the neutral value. If the assumption of heterogeneity was not met, a common effects model was applied. Prior to data extraction, a prespecified subgroup analysis was formulated, which included the following: (i) studies performed prior to 2020 vs. studies performed after 2020, (ii) gender (male vs. female), (iii) follow-up duration (<three months vs. ≥three months), (iv) endotype (VSA vs. MVA) and (v) type of study (RCT vs. cohort studies). The subgroup analysis was only undertaken if two or more studies were available for each arm of the proposed subgroups. The aim was to determine the effect of possible effect modifiers on the primary outcome. In addition, a prespecified sensitivity analysis was formulated with the aim of determining the effect of the variety in the overall risk of bias for the studies on the primary outcome by performing stratified analyses: one based on all studies included, one based on studies at low risk and one based on studies at high risk of bias. Publication bias assessment was determined using the Egger’s test and funnel plot for visual assessment. Statistical analysis was performed using R 2023.06.2.

### 2.7. Confidence in Cumulative Evidence

The overall quality of evidence of the systematic review was assessed by two independent reviewers (F.V. and I.W.) using the Grading of Recommendations Assessment Development and Evaluation (GRADE) tool. The GRADE framework consists of five domains: (i) study limitations, (ii) inconsistency in results, (iii) indirectness of evidence, (iv) imprecision and (v) publication bias. At the start of the GRADE assessment, the evidence was assumed to be high quality and was downgraded based on any of the five domains to moderate, low or very low quality of evidence.

## 3. Results

### 3.1. Literature Search and Study Characteristics

In total, 13,803 citations were identified, and, after screening, a total of 425 full-text articles were evaluated (Figure 1). For the final analysis, 125 studies were included.

Eighty studies were incorporated into meta-analysis, with the remainder (45 studies) synthesised narratively. Study characteristics of the 80 studies are summarised in Table 2 according to treatment modality (Appendix A for the remaining 45 studies) [2,13,14,15,16,17,18,19,20,21,22,23,24,25,26,27,28,29,30,31,32,33,34,35,36,37,38,39,40,41,42,43,44,45,46,47,48,49,50,51,52,53,54,55,56,57,58,59,60,61,62,63,64,65,66,67,68,69,70,71,72,73,74,75,76,77,78,79,80,81,82,83,84,85,86,87,88,89,90,91,92].

In summary, (I) sixteen studies regarding calcium channel blockers, (II) eleven studies regarding lifestyle interventions, (III) ten studies regarding RAAS inhibitors, (IV) ten studies regarding beta-blockers, (V) eight studies regarding long-acting nitrates, (VI) seven studies regarding statins, (VII) six studies regarding neuromodulation, (VIII) six studies regarding ranolazine, (IX) five studies regarding trimetazidine, (X) five studies regarding traditional Chinese medicine, (XI) four studies regarding hormone therapy, (XII) two studies regarding stem cell therapy and (XIII) three studies regarding enhanced external counterpulsation were included. Of all included studies, 10 studies met the current ESC 2020 guideline definition of ANOCA, and 51 out of 80 studies (cross-over studies are only counted once in this count; see Table 2) were randomised controlled trials. Table 3 gives an overview of the endpoints available per treatment modality.

### 3.2. Primary Outcome

In total, 37 studies determined the effect of 11 treatment modalities on angina pectoris (AP) frequency. The strongest reduction in AP frequency was found in treatment with neuromodulation, with an estimated −3.35 difference in standardised mean difference (SMD) (95% CI: −5.13; −1.56) (random effects model (REM)). Furthermore, a statistically significant reduction in AP frequency was found for the treatment modalities trimetazidine (−1.74 SMD, 95% CI: −2.63; −0.85) (REM), traditional Chinese medicine (−1.55 SMD, 95% CI: −2.36; −0.75) (REM), beta-blockers (−1.32 SMD, 95% CI: −1.88; −0.77) (common effects model (CEM)), enhanced external counterpulsation (−1.27 SMD, 95% CI: −2.04; −0.49) (REM), stem cell therapy (−1.04 SMD, 95% CI: −1.51; −0.57) (CEM), lifestyle interventions (−0.86 SMD, 95% CI: −1.15; −0.57) (CEM), RAAS inhibitors (−0.83 SMD, 95% CI: −1.31; −0.35) (REM) and calcium channel blockers (−0.64 SMD, 95% CI: −0.92; −0.35) (CEM). No statistically significant effect was found in the treatment modalities ranolazine (−0.20 SMD, 95% CI: −0.64; 0.24) (CEM) or long-acting nitrates (−0.03 SMD, 95% CI: −0.46; 0.39) (CEM) (Figure 2).

### 3.3. Secondary Outcomes

Fifty studies looked at the effect of 11 treatment modalities on changes in exercise capacity, and the largest improvement was seen in treatment with statins, with an estimated 0.90 SMD (95% CI: 0.43; 1.37) (CEM). Other treatment modalities that showed a statistically significant improvement were lifestyle interventions (0.79 SMD, 95% CI: 0.44; 1.15) (REM), traditional Chinese medicine (0.75 SMD, 95% CI: 0.41; 1.09) (CEM), neuromodulation (0.72 SMD, 95% CI: 0.32; 1.11) (CEM) and RAAS inhibitors (0.57 SMD, 95% CI 0.27; 0.87) (CEM) (Figure 3). Eighteen studies determined the quality of life outcome measure for seven treatment modalities, where the biggest improvement was seen in treatment with neuromodulation (2.29 SMD, 95% CI: 0.61; 3.97) (REM), with statistically significant change also seen in lifestyle interventions (0.99 SMD, 95% CI: 0.36; 1.63) (REM) and stem cell therapy (0.85 SMD, 95% CI: 0.38; 1.32) (CEM) (Figure 4). For the remaining secondary outcomes (CCS class, changes in CFR and survival), two statistically significant changes were seen in CCS class for the treatment modalities stem cell therapy (−1.61, 95% CI: −2.44; −0.78) (REM) and enhanced external counterpulsation (−1.60, 95% CI: −2.71; −0.48) (REM). The remainder of the treatment modalities did not show a pooled statistically significant effect (Table 3, Appendix A).

### 3.4. Publication Bias Assessment

Egger’s test for asymmetry was performed for each outcome to determine publication bias, and in the outcomes angina frequency (*p* = 0.0230), quality of life (*p* < 0.0001) and coronary blood flow (*p* = 0.0167), the test was statistically significant. For the outcome exercise capacity, the Egger’s test was not statistically significant (*p* = 0.1127), and for the remaining outcomes (CCS class and survival), there were too few studies, and the Egger’s test could not be performed. In addition, a funnel plot was made for all outcome parameters, and taking into consideration the visual assessment of the funnel plots publication, bias was present for all reported outcomes (Appendix A).

### 3.5. Subgroup Analysis

There were five prespecified subgroup analyses described prior to starting the systematic review, and based on the results gathered, three out of five could be performed. Two of the five could not performed due to lack of sufficient information gathered from the individual studies; this was with regard to the subgroup analysis based on type of ANOCA (VSA vs. MVA) and gender. The other prespecified subgroup analyses were performed in those cases in which two or more studies could be included in a subgroup for the analysis looking at the primary outcome. In the subgroup analysis assessing studies published prior to 2020 vs. after 2020, no significant differences were found (Appendix A). With regard to the effect of differences in follow-up duration between studies, one statistically significant difference was seen in the treatment with neuromodulation in angina pectoris frequency, with −5.00 SMD (−7.38; −2.61) for <3 months vs. −1.42 SMD (−3.13; 0.30) for ≥3 months (*p* = 0.0169) (Appendix A). The subgroup analysis assessing differences in type of study performed yielded no statistically significant differences (Appendix A).

### 3.6. Sensitivity Analysis

To determine the effect of the variety in the overall risk of bias for the studies on the primary outcome (angina pectoris frequency), all studies were assessed. Thirty-six studies were included in the meta-analysis for the primary outcome and were assessed for overall risk of bias (Table 4). Only three RCTs had an overall low risk of bias, and seven cohort studies scored ‘Good’. No sensitivity analysis could be performed because too few studies that looked at different treatment modalities had an overall low risk of bias.

### 3.7. Confidence in Cumulative Evidence

The overall confidence in the cumulative evidence was assessed using the Grading of Recommendations Assessment Development and Evaluation (GRADE) tool. Based on the five elements of the GRADE tool, there is (i) a significant risk of bias based on the risk of bias assessment performed (Table 4). (ii) There is a certain amount of imprecision when viewing the 95% confidence intervals of the outcomes; imprecision is lowest for the primary outcome (Figure 2, Figure 3 and Figure 4, Appendix A). (iii) There is some inconsistency in the results; this varies per treatment modality and per outcome measure. Looking specifically at the primary outcome, the inconsistency of results appears to be limited. (iv) Due to the variation in the definition used for patients with ANOCA in the included studies, there is a large possibility of indirectness of evidence. Only six studies used the current ANOCA definition. (v) Based on the analysis performed, there is publication bias present. Taking the five elements into consideration, the overall confidence in the cumulative evidence is very low.

## 4. Discussion

This comprehensive systematic review and meta-analysis, which aimed to include all studies (randomised and non-randomised) performed up to November 2024 regarding treatment modalities for patients with ANOCA, shows significant improvement in angina pectoris frequency for the vast majority of the treatment modalities, including neuromodulation, trimetazidine, traditional Chinese medicine, beta-blockers, enhanced external counterpulsation, stem cell therapy, lifestyle interventions, RAAS inhibitors and calcium channel blockers.

With this systematic review, we have attempted to provide an overview of all studies performed to date that have looked at possible treatments for patients with ANOCA and to assess their pooled effect on various outcomes with meta-analyses. One prior systematic review has also attempted to provide these insights but was unable to perform a meta-analysis due to very strict inclusion criteria [93]. The current meta-analysis has shown that a large number of treatment modalities have already been researched for this patient population, but that patient numbers are relatively small. It provides a base of current knowledge and can be used to help guide and perform future trials for those treatment modalities that appear to have a positive effect on the endpoints provided.

The definition of ANOCA was coined and published in 2020 in the ESC consensus document on ischemia with non-obstructive coronary arteries [3]. Prior to 2020, there was no uniform definition for patients with ANOCA, and the more general term ‘cardiac syndrome X’ (CSX) was used. There are multiple definitions of cardiac syndrome X, ranging from angina pectoris and normal coronary arteries with no additional prove of myocardial ischaemia required through to angina pectoris, normal coronary arteries and a positive SPECT [94]. A number of included studies used a positive exercise test as inclusion criterion, whilst it is known that an exercise test has limited specificity in detecting CAD [7]. This broad spectrum of definitions of CSX has possibly led to inclusion of patients in whom the cause of symptoms was not cardiac in origin [95]. The majority of studies analysed in the meta-analysis were performed and published prior to 2020 and used the cardiac syndrome X definition, probably resulting in a more heterogenous patient population in those studies. It is unclear how many patients included in these studies would have met the current ANOCA criteria and whether the same effects would have been found. It could be postulated that a more selective patient population would lead to a larger effect, as these patients have a proven cardiac origin of symptoms, which is the target of the investigated treatment modalities.

During the initial set-up of the meta-analysis an aim was to perform a network meta-analysis. This was not possible due to too much heterogeneity in the study populations of the various studies, underlining the importance of a uniform definition. A previous systematic review used a more stringent set of inclusion criteria but found too few articles to be able to perform a meta-analysis [93]. In this systematic review, an expressed choice was made to use a broader set of inclusion criteria, with the aim of gathering as much current data as possible, accepting the more heterogeneous patient population.

There was variety amongst the included studies in their study protocol with regard to (dis)continuation of concomitant treatments. It is known that patients with persisting angina pectoris have ‘optimal medical therapy’, which usually includes a combination of multiple pharmacological agents such as beta-blockers, calcium channel blockers and long-acting nitrates [96]. The included studies all looked at one or two (cross-over design) specific treatment modalities but varied in their approach to concomitant therapies. Some studies allowed continuation of other anti-anginal medication, whilst other studies had patients stop certain anti-anginal medications that were not allowed during the trial period. The concomitant use of other (anti-anginal) therapies can have a significant impact on the results found in the studies that did not clearly discontinue concomitant anti-anginal therapies, possibly leading to a smaller and/or non-significant effect.

The follow-up duration in the studies ranged from a few days [26] up to several years [63]. For clinicians, it is important to know what the effect of a treatment modality is both short- and long-term. A subgroup analysis was performed to determine if the found effects were dependent on the follow-up duration. For the primary endpoint, angina pectoris frequency, there was a statistically significant difference in effect for neuromodulation, with less effect at long-term follow-up (defined as ≥3 months). This could suggest attenuation to that specific treatment modality, a phenomenon that is known to occur in patients using long-acting nitrates, although this was not seen in the current meta-analysis [97]. These findings should be cautiously interpreted, as the subgroup analysis included a small number of studies, and it was only seen for one (of the twelve) treatment modalities.

The majority of studies chose clinical endpoints such as angina pectoris frequency and exercise capacity to assess treatment effect. A very limited amount of data was available on the effect of the various treatment modalities on the coronary blood flow (CBF). This is a hiatus because it is very valuable to gather more data on the influence of the treatments on CBF, as this is the base for the diagnosis of ANOCA. The secondary outcome, CBF, included only four treatment modalities with small patient numbers, and none of the effects found were statistically significant. With the expanding knowledge regarding the underlying pathophysiology of ANOCA (both endotypes, MVA and VSA), it is relevant to gain more insight into the effects of possible therapies on these underlying pathophysiological mechanisms. This could lead to more targeted therapies based on the specific endotype, which has proven to be effective in previous studies [98,99]. It could also provide clinicians with a tangible way to measure whether the chosen therapy has the desired effect by repeating CFT after initiating treatment and modifying the treatment based on CFT findings.

The outcomes of this meta-analysis were reported in standardised mean difference (SMD) because the studies used various methods to determine outcomes. Initially, each SMD was to be transformed back into the scale that was most frequently used in the included studies to allow better clinical interpretation of the effect. This can be achieved by multiplying the SMD by an estimate of the SD associated with the most frequently used scale [9]. To obtain the SD, a weighted average across all intervention groups of all studies that used the selected instrument should be calculated [9]. During the analysis of the included studies in this meta-analysis, it was seen that for each treatment modality and for each outcome, the most frequently used scale varied frequently per treatment modality and per outcome. This variability made it impossible to use one scale for each outcome. To retain uniformity, it was decided by the research team to report the SMD instead of the transformed value. An important limitation of reporting the SMD is the clinical interpretability of the value. However, by reporting the SMD and retaining uniformity per outcome, clinicians can compare the effect sizes of each treatment modality to the other treatment modalities, showing which treatment modality appears to have the largest effect for each outcome.

The robustness of the results found in this meta-analysis is dependent on the presence of publication bias, the risk of bias assessment of the individual studies and the overall GRADE evaluation performed. Publication bias was present for all outcomes, which could lead to an overestimation of the found pooled effects, and this should be taken into consideration when interpreting the found results. With regard to the risk of bias of the individual studies, this was only performed for the studies which looked at the primary outcome (angina pectoris frequency), and the majority of studies had ‘some concerns’ or ‘high risk’ of overall risk of bias adding to the uncertainty of the robustness of the found results. Both these limitations are reflected in the very low GRADE level of evidence reported. Clinicians should be aware of the very low GRADE level of evidence when interpreting the found results and recognise that the pooled effects found could be an overestimation.

All reported outcomes had a positive trend, although approximately 50% of the results found were not statistically significant. Important contributing factors are the small numbers of patients included in the individual studies and the heterogeneity amongst the included studies. The gaps of evidence, based on the results of this meta-analysis, are the largest for long-acting nitrates and ranolazine, with small patient numbers and no significant pooled effects, whilst the most evidence is currently present for calcium channel blockers and lifestyle interventions. The results show the importance of performing additional randomised controlled trials in which the ANOCA criteria are used, thus creating a more homogenous patient population. This will lead to additional robust results that will help guide clinicians in the treatment of a large and growing patient population in which, currently, up to 25% remain symptomatic despite ‘optimal medical therapy’ [3].

## 5. Conclusions

This systematic review and meta-analysis into the various treatment modalities for patients with ANOCA provides an overview showing improvement of angina pectoris frequency and exercise capacity for a majority of the studied treatment modalities. Important limitations are the fact that that only ten studies meet the current definition of ANOCA and the heterogeneity of the studied treatment modalities on the reported outcomes. This review underlines the importance of undertaking new randomised controlled trials with existing treatment modalities to determine the efficacy in patients with angina pectoris meeting the criteria for ANOCA.

## Figures and Tables

**Figure 1 jcm-14-04069-f001:**
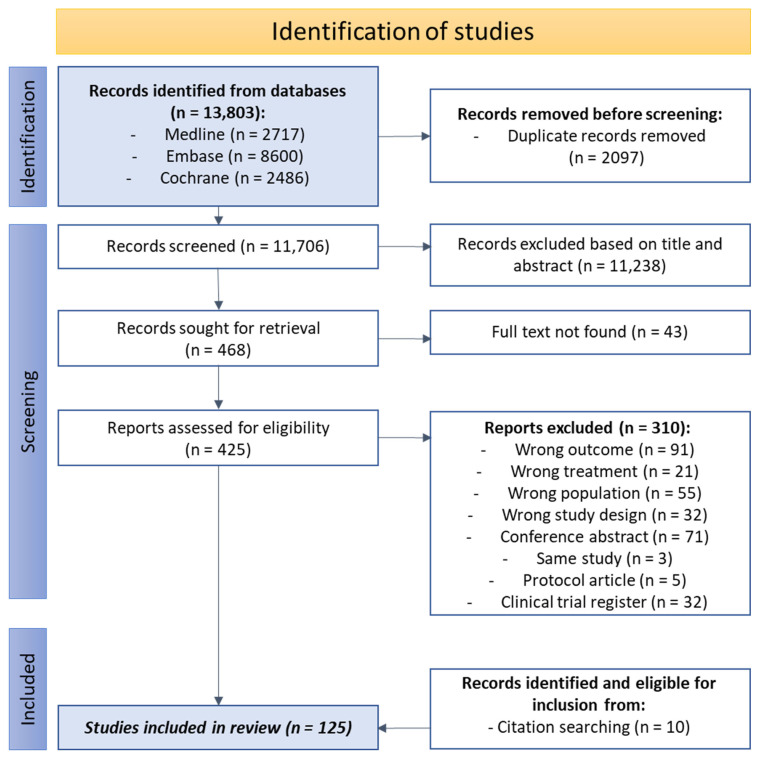
PRISMA flowchart.

**Figure 2 jcm-14-04069-f002:**
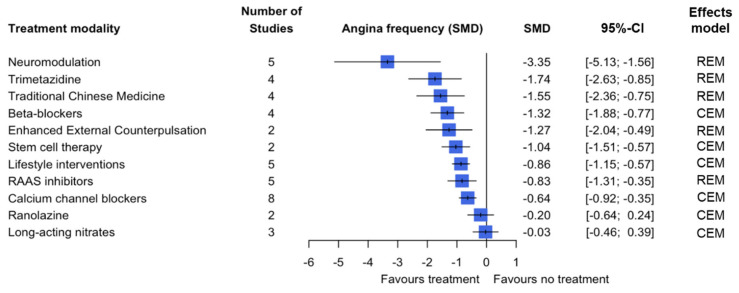
Pooled estimate of the treatment effect on angina pectoris frequency per treatment modality.

**Figure 3 jcm-14-04069-f003:**
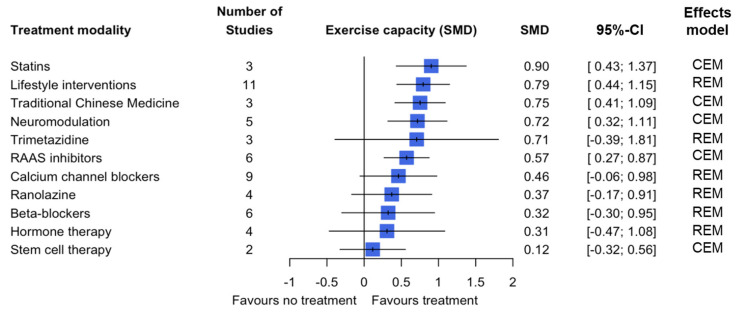
Pooled estimate of the treatment effect on exercise capacity per treatment modality.

**Figure 4 jcm-14-04069-f004:**
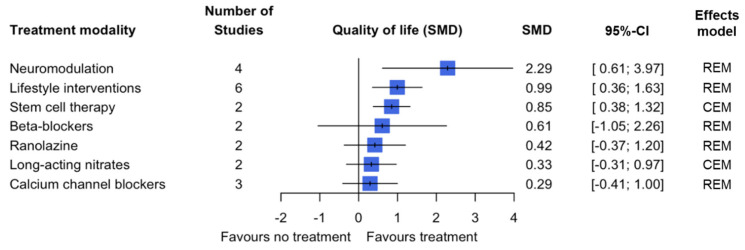
Pooled estimate of the treatment effect on quality of life per treatment modality.

**Table 1 jcm-14-04069-t001:** Definitions for search strategy.

*Angina with Non-Obstructive Coronary Arteries (ANOCA)*, according to endotype. - Microvascular angina (MVA): ○ Symptoms of myocardial ischaemia; ○ Absence of obstructive CAD (<50% diameter reduction or FFR > 0.80); ○ Objective evidence of myocardial ischaemia; ○ Evidence of impaired coronary microvascular function; CFR < 2.0 and/or coronary microvascular spasm (angina, ischaemic ECG changes but no epicardial spasm during acetylcholine testing) and/or IMR ≥ 25). - Vasospastic angina (VSA): ○ Nitrate responsive angina; ○ Transient ischaemic ECG changes; ○ Coronary artery spasm: transient total or subtotal (>90%) coronary artery occlusion with angina and ischaemic ECG changes spontaneously or in response to acetylcholine.
*‘normal coronary arteries and angina’*: alternative terms used: - Angina X syndrome - Cardiac syndrome X - Angina pectoris with normal coronary arteriogram
*Treatment modalities and outcomes*: all treatments, including pharmacological and non-pharmacological, used in patients with ANOCA or ‘normal coronary arteries and angina’ with the aim of symptom reduction and/or improvement in quality of life and/or changes in myocardial blood flow.

**Table 2 jcm-14-04069-t002:** Study characteristics.

First Author	Year	Study Type	Follow-up Duration	N	Mean Age Intervention	Mean Age Control	Male (%)	Diagnosis ANOCA
** *Calcium channel blockers* **				** *538* **				
Sinha [13]	**2024**	Prospective, randomised	12 weeks	87	62 ± 8	60 ± 7	37	MVA—typical angina, preserved left ventricular ejection fraction (>50%) and normal CAG (FFR > 0.8)
Jansen [2]	**2022**	Prospective, randomised	6 weeks	85	57.7 ± 8.8	58.0 ± 9.3	34	VSA/MVA—typical angina and CFT; CFR ≤ 2.0 and/or IMR > 25 and/or abnormal response to Ach
Kook [14]	**2020**	Prospective, randomised	12 weeks	48	59.5 ± 11.8	62.8 ± 7.2	66.7	VSA—typical angina and CAG with abnormal response to Ach
Zhang [15]	2014	Prospective, randomised	90 days	66	54 ± 7	53 ± 8	45.5	Cardiac syndrome X—typical angina, positive exercise test and normal CAG
Oikawa [16]	2010	Prospective, randomised	8 weeks	28	64.6 ± 10.8	61.2 ± 14	78.6	VSA—typical angina and CAG with abnormal response to Ach
Özçelik [17]	1999	Prospective, randomised (cross-over)	12 weeks	18	46 ± 10	N/A	38.9	MVA—typical angina, positive exercise test, negative IV ergonovine test and normal CAG
Lanza [18]	1999	Prospective, randomised (cross-over)	16 weeks	10	57 ± 6	N/A	40	Cardiac syndrome X—typical angina, positive exercise test and normal CAG
Vogt [19]	1994	Prospective, unrandomised	52 weeks	15	61 ± 7	N/A	66.7	Cardiac syndrome X—typical angina, positive exercise test or MPS with ischaemia and normal CAG
Cannon [20]	1990	Prospective, randomised (cross-over)	4 weeks	22	52 [30–65]	N/A	45.5	MVA—typical angina, normal CAG and reduced CFR
Romeo [21]	1988	Prospective, randomised (cross-over)	9 weeks	30	50 ± 9	N/A	10	MVA—typical angina, positive exercise test, negative IV ergonovine test and normal CAG
Prida [22]	1987	Prospective, randomised (cross-over)	16 weeks	15	58.3 ± 10.5	N/A	80	VSA—typical angina and CAG with spontaneous or Ach-induced spasm
Kugiyama [23]	1986	Prospective, randomised (cross-over)	3 weeks	20	54.2 ± 7.8	N/A	80	VSA—typical angina, CAG with proven coronary spasm and normal coronary arteries
Gelman [24]	1985	Prospective, randomised (cross-over)	4 weeks	17	57.2 ± 6.15	N/A	94.1	VSA—typical angina and CAG with spontaneous or Ach-induced spasm
Cannon [25]	1985	Prospective, randomised (cross-over)	4 weeks	26	53 [38–64]	N/A	42.3	Cardiac syndrome X—typical angina, normal CAG and abnormal vasodilator reserve
Pitcher [26]	1981	Prospective, randomised (cross-over)	4 weeks	33	49 [31–58]	N/A	27.3	Cardiac syndrome X—typical angina, positive exercise test and normal CAG
Freedman [27]	1981	Prospective, unrandomised	4 days	6	56 [49–64]	N/A	83.3	Cardiac syndrome X—typical angina, positive exercise test and normal CAG
** *Lifestyle interventions* **				** *389* **				
Sugisawa [28]	**2021**	Prospective, randomised	12 weeks	20	58.1 ± 2.3	61.8 ± 3.2	25	VSA—typical angina and CAG with abnormal response to Ach
Bove [29]	**2020**	Prospective, randomised	24 weeks	56	64.3 ± 7.6	63.0 ± 8.0	0	MVA—typical angina and CFT; CFR ≤ 2.5
Rahmani [30]	2020	Prospective, randomised	4 weeks	30	53 ± 9	54 ± 7	20	Cardiac syndrome X—typical angina, positive exercise test and/or MPS with ischaemia and normal CAG
Szot [31]	2016	Prospective, unrandomised	12 weeks	55	57.3 ± 5.4	N/A	0	Cardiac syndrome X—typical angina, MPS with ischaemia and normal CAG
de Carvalho [32]	2015	Prospective, unrandomised	16 weeks	12	53.8 ± 9.7	N/A	58.3	Cardiac syndrome X—typical angina, MPS with ischaemia and normal CAG
Feizi [33]	2012	Prospective, randomised	8 weeks	40	50.5 ± 7.1	52.4 ± 6.3	0	Cardiac syndrome X—typical angina, MPS with ischaemia and normal CAG
Asbury [34]	2009	Prospective, randomised	16 weeks	53	58.1 ± 7.2	56.1 ± 8.6	0	Cardiac syndrome X—typical angina, positive exercise test and normal CAG
Asbury [35]	2008	Prospective, randomised	8 weeks	64	58.1 ± 9.4	56.4 ± 7.8	0	Cardiac syndrome X—typical angina, positive exercise test and normal CAG
Tyni-Lenne [36]	2002	Prospective, randomised	8 weeks	24	57 ± 7	55 ± 8	0	Cardiac syndrome X—typical angina, positive exercise test and normal CAG
Cunningham [37]	2000	Prospective, unrandomised	12 weeks	9	56 [48–66]	N/A	0	Cardiac syndrome X—typical angina, positive exercise test and normal CAG
Eriksson [38]	2000	Prospective, randomised	16 weeks	26	57 ± 7	53 ± 10	0	Cardiac syndrome X—typical angina, positive exercise test and normal CAG
** *RAAS inhibitors* **				** *339* **				
Michelsen [39]	2018	Prospective, randomised	24 weeks	63	58.6 ± 11.6	57.3 ± 12.5	0	MVA—typical angina, CAG with no epicardial stenosis > 50% and CFVR < 2.2 (adenosine stress echocardiography)
Bavry [40]	2014	Prospective, randomised	16 weeks	51	54 ± 10	54 ± 11	0	MVA—typical angina, CAG with no epicardial stenosis > 50% and endothelial dysfunction (<5% diameter increase Ach)
Pauly [41]	2011	Prospective, randomised	16 weeks	61	56 ± 8	51 ± 10	0	Cardiac syndrome X—typical angina, CAG with no epicardial stenosis > 50% and CFR < 3.0
Pizzi [42]	2004	Prospective, randomised	24 weeks	45	59.6 ± 8.7	57.6 ± 9.6	11.1	Cardiac syndrome X—typical angina, positive exercise test, normal CAG and no coronary spasm during ergonovine IV
Chen [43]	2002	Prospective, randomised	8 weeks	20	66.3 ± 3.5	67.7 ± 2.9	75	Cardiac syndrome X—typical angina, positive exercise test, normal CAG and no evidence of coronary spasm
Kanadaşi [44]	2002	Prospective, randomised (cross-over)	16 weeks	21	49.5 ± 10.4	N/A	14.3	Cardiac syndrome X—typical angina, positive exercise test and normal CAG
* Özçelik [17] *	* 1999 *	* See previous * *						
Nalbantgil [45]	1998	Prospective, randomised (cross-over)	10 weeks	35	43.9 ± 6.4	N/A	22.9	MVA—typical angina, positive exercise test and normal CAG
Motz [46]	1996	Prospective, unrandomised	12 weeks	15	58 ± 6	N/A	66.7	Cardiac syndrome X—typical angina, positive exercise test and normal CAG
Kaski [47]	1994	Prospective, randomised (cross-over)	4 weeks	10	53 ± 6	N/A	30	Cardiac syndrome X—typical angina, positive exercise test, abnormal coronary flow reserve and normal CAG
** *Beta-blockers* **				** *219* **				
* Kook [14] *	* 2020 *	* See previous * *						
Erdamar [48]	2009	Prospective, randomised	12 weeks	30	47.6 ± 7.2	49.1 ± 7.3	43.3	Cardiac syndrome X—typical angina, positive exercise test, normal CAG and absence of coronary spasm
Sen [49]	2009	Prospective, randomised	12 weeks	34	47.2 ± 7.3	49.5 ± 7.3	70.6	Cardiac syndrome X—typical angina, positive exercise test, normal CAG and absence of coronary spasm
Suzuki [50]	2003	Prospective, unrandomised	12 weeks	12	56.3 ± 8.2	N/A	58.3	VSA—typical angina and CAG with abnormal response to Ach
* Kanadaşi [44] *	* 2002 *	* See previous * *						
* Lanza [18] *	* 1999 *	* See previous * *						
Leonardo [51]	1999	Prospective, randomised (cross-over)	8 weeks	16	62 ± 7	N/A	18.8	Cardiac syndrome X—typical angina, positive exercise test and normal CAG
Shimizu [52]	1993	Prospective, randomised (cross-over)	1 week	10	57.5 ± 6.7	N/A	100	VSA—typical angina, CAG with no epicardial stenosis > 50% and spontaneous or Ach-induced coronary spasm
* Romeo [21] *	* 1988 *	* See previous * *						
* Kugiyama [23] *	* 1986 *	* See previous * *						
** *Long-acting nitrates* **				** *2792* **				
Lim [53]	2022	Prospective, unrandomised	24 months	568	54.9 ± 11.3	55.6 ± 11.5	55.5	VSA—typical angina, normal CAG and positive ergonovine provocation test
Kim [54]	2018	Prospective, unrandomised	54.7 months	1127	56.7 ± 9.3	56.6 ± 9.8	85.5	VSA—typical angina, normal CAG and positive ergonovine provocation test
Takahashi [55]	2015	Prospective, unrandomised	32 months	826	66 [58–73]	66 [59,60,61,62,63,64,65,66,67,68,69,70,71,72,73]	74.8	VSA—typical angina, CAG with no epicardial stenosis > 50% and positive ergonovine/Ach provocation test
Wu [56]	2015	Prospective, randomised (cross-over)	4 weeks	9	59 ± 9	N/A	22.2	MVA—typical angina, normal CAG, positive exercise test and CFR < 2.0 (Doppler LAD)
Kosugi [57]	2011	Prospective, unrandomised	70.5 months	231	61.0 ± 10.6	59.2 ± 9.9	66.7	VSA—typical angina, normal CAG and positive Ach provocation test
* Kanadaşi [44] *	* 2002 *	* See previous * *						
* Lanza [18] *	* 1999 *	* See previous * *						
** *Statins* **				** *7479* **				
Lee [58]	**2024**	Prospective, unrandomised	4.8 years	4432	57.8 ± 11.6	58.5 ± 13.1	45.6	VSA—typical angina, normal CAG and positive ergonovine/Ach provocation test
Mori [59]	2022	Prospective, unrandomised	726 days	422	65.5 ± 9.5	64.6 ± 10.3	74.4	VSA—typical angina, CAG with no epicardial stenosis > 50% and positive ergonovine/Ach provocation test
Seo [60]	2020	Prospective, unrandomised	104 weeks	1658	55.9 ± 10.9	53.5 ± 11.5	60.6	VSA—typical angina, normal CAG and positive ergonovine provocation test
Ishii [61]	2016	Prospective, unrandomised	60 months	256	64.6 ± 9.9	64.8 ± 9.7	43.8	VSA—typical angina, normal CAG and positive Ach provocation test
Oh [62]	2016	Prospective, unrandomised	4.5 years	562	55.8 ± 9.2	55.7 ± 9.2	85.2	VSA—typical angina, normal CAG and positive ergonovine provocation test
* Zhang [15] *	* 2014 *	* See previous * *						
* Pizzi [42] *	* 2004 *	* See previous * *						
Kayikciolgu [63]	2003	Prospective, randomised	12 weeks	38	45 ± 7	47 ± 4	42.1	Cardiac syndrome X—typical angina, positive exercise test and normal CAG
** *Neuromodulation* **				** *77* **				
de Vries [64]	2007	Prospective, unrandomised	5.1 years	12	56.7 ± 8.2	N/A	37.5	Cardiac syndrome X—typical angina and normal CAG
Sgueglia [65]	2007	Prospective, unrandomised	36 months	28	60.9 ± 8.5	60.9 ± 8.8	28.6	Cardiac syndrome X—typical angina, positive exercise test OR perfusion defect MPS and normal CAG
Jessurun [66]	2003	Prospective, unrandomised	4 weeks	8	55 ± 7	N/A	37.5	Cardiac syndrome X—typical angina, normal CAG and heterogeneous myocardial perfusion MPS
Lanza [67]	2005	Prospective, randomised (cross-over)	7 weeks	10	58.6 ± 5.7	N/A	30	Cardiac syndrome X—typical angina, positive exercise test OR perfusion defect MPS and normal CAG
Lanza [68]	2001	Prospective, unrandomised	1 month	7	59.3 ± 11	N/A	57.1	Cardiac syndrome X—typical angina, positive exercise test and normal CAG
Eliasson [69]	1993	Prospective, unrandomised	1 week	12	61 ± 6	N/A	33.3	Cardiac syndrome X—typical angina, positive exercise test and normal CAG
** *Ranolazine* **				** *246* **				
* Sinha [13] *	* 2024 *	* See previous * *						
Birkeland [70]	2017	Prospective, randomised (cross-over)	2 weeks	30	54 ± 10.6	N/A	3.3	MVA—typical angina, CAG with no epicardial stenosis > 50% and CFR < 2.5 OR MPRI < 2.0
Ahmed [71]	2017	Prospective, unrandomised	4 weeks	7	57.6 ± 11.3	N/A	57.1	MVA—typical angina, CAG with no epicardial stenosis > 50%, positive exercise test OR perfusion defect MPS OR stress echo with RWMA and IMR > 20
Safdar [72]	2017	Prospective, randomised	4 weeks	31	50 ± 5	50 ± 7	29	MVA—typical angina, normal CAG and CFR < 2.5
Merz [73]	2016	Prospective, randomised (cross-over)	2 weeks	132	55.2 ± 9.8	N/A	4	MVA—typical angina, CAG with no epicardial stenosis > 50%, CFR < 2.5 OR no dilation with Ach OR MPRI < 2.0
Villano [74]	2013	Prospective, randomised	4 weeks	46	57 ± 11	60 ± 9	19.6	MVA—typical angina, positive exercise test, normal CAG, CFR < 2.5 (Doppler LAD) and no vasospastic angina
** *Trimetazidine* **				** *195* **				
Boldueva [75]	2020	Prospective, randomised	3 months	60	58.4 ± 6.5	57.3 ± 6.4	43.3	MVA—typical angina, positive exercise test, normal CAG and ischaemia using PET
Galin [76]	2016	Prospective, unrandomsed	6 months	50	55.2 ± 3.8	N/A	32	Cardiac syndrome X—typical angina, CAG with no epicardial stenosis > 50% and positive exercise test
Rogacka [77]	2000	Prospective, unrandomised	6 months	34	46 [32–60]	N/A	41.2	Cardiac syndrome X—typical angina, positive exercise test and normal CAG
* Leonardo [51] *	* 1999 *	* See previous * *						
Nalbantgil [78]	1999	Prospective, randomised (cross-over)	10 weeks	35	43.9 ± 6.4	N/A	22.9	Cardiac syndrome X—typical angina, positive exercise test and normal CAG
** *Traditional Chinese medicine* **				** *356* **				
Noroozi [79]	2023	Prospective, unrandomised	90 days	28	50.6 ± 6	N/A	42.9	Cardiac syndrome X—typical angina, positive exercise test and normal CAG
Ma [80]	2021	Prospective, randomised	12 weeks	171	60.2 ± 6.2	59.1 ± 6.2	Not repor.	Cardiac syndrome X—typical angina, positive exercise test, normal CAG and negative ergonovine test
Cao [81]	2021	Prospective, randomised	Not repor.	70	60.6 ± 10	61.9 ± 9.3	56.9	Cardiac syndrome X—typical angina, positive exercise test and normal CAG
Li [82]	2007	Prospective, randomised	3 months	36	Not repor.	Not repor.	Not repor.	Cardiac syndrome X—typical angina, positive exercise test and normal CAG
Mao [83]	2007	Prospective, unrandomised	14 days	51	51.2 ± 6.2	50.8 ± 6.5	21.6	Cardiac syndrome X—typical angina, positive exercise test and normal CAG
** *Hormone therapy* **				** *110* **				
Merz [84]	2010	Prospective, randomised	12 weeks	37	56 ± 9	59 ± 7	0	Cardiac syndrome X—typical angina, CAG with no epicardial stenosis > 50% and positive exercise test OR perfusion defect MPS OR CFR < 2.25
Adamson [85]	2001	Prospective, randomised (cross-over)	16 weeks	32	58 ± 2	N/A	0	Cardiac syndrome X—typical angina, positive exercise test and normal CAG
Rosano [86]	1996	Prospective, randomised (cross-over)	18 weeks	26	56.8 [47–65]	N/A	0	Cardiac syndrome X—typical angina, positive exercise test and normal CAG
Albertsson [87]	1996	Prospective, randomised (cross-over)	1 week	15	58 ± 6	N/A	0	Cardiac syndrome X—typical angina, positive exercise test and normal CAG
** *Autologous CD34+ stem cell therapy* **				** *40* **				
Henry [88]	**2022**	Prospective, unrandomised	6 months	20	54.3 ± 12.7	N/A	15	MVA—typical angina, CAG with no epicardial stenosis > 40% and CFR ≤ 2.5
Corban [89]	**2022**	Prospective, unrandomised	6 months	20	51.0 ± 12.1	N/A	25	MVA—typical angina, CAG with no epicardial stenosis > 40% and CFR ≤ 2.5
** *Enhanced External Counterpulsation* **				** *181* **				
Ashokprabhu [90]	**2024**	Retrospective, unrandomised	7 weeks	101	60.6 ± 11.3	N/A	37.6	ANOCA—CCS class III or IV and absence of obstructive coronary arteries (CAG or CCTA stenosis < 50%)
Zhang [91]	**2024**	Prospective, randomised	1 year	80	50.5 ± 16.8	51.2 ± 14.6	67.5	MVA—typical angina, MPR < 2.0 (CMR) and absence of obstructive coronary arteries (CAG or CCTA stenosis < 50%)
Kronhaus [92]	2009	Prospective, unrandomised	12 months	30	64.9 ± 10.7	N/A	27	Cardiac syndrome X—typical angina, no obstructive coronary arteries (<50%) and pharmacological or exercise-induced ischaemia.

ANOCA: angina with no obstructive coronary arteries, VSA: vasospastic angina, MVA: microvascular angina, CFT: coronary function testing, CFR: coronary flow reserve, IMR: index of microvascular resistance, CAG: coronary angiogram, N/A: not applicable, IV: intravenous, MPS: myocardial perfusion scan, Ach: acetylcholine, CFVR: coronary flow velocity reserve, LAD: left anterior descending, MPRI: myocardial perfusion reserve index, RWMA: regional wall motion abnormalities, PET: positron emission tomography, CCTA: coronary computed tomography angiography, MPR: myocardial perfusion reserve, CMR: cardiac magnetic resonance. **Bold**: study meets the ESC 2020 guideline definition of ANOCA. * Cross-over studies in which participants received multiple treatment modalities vs. placebo in sequence with wash-out period prior to switching to the next treatment modality; hence, the number of patients and patient characteristics are the same for each treatment modality.

**Table 3 jcm-14-04069-t003:** Overview of primary and secondary endpoints per treatment modality and statistical significance. CCS = Canadian Cardiovascular Society, EECP = enhanced external counterpulsation, n = number of patients. **Green** = statistically significant result, **Orange** = not statistically significant result, **Red** = no result, not enough studies available for a meta-analysis.

	Angina Pectoris Frequency	Exercise Capacity	Quality of Life	CCS Class	Coronary Blood Flow	Survival
Calcium channel blockers	n = 188	n = 223	n = 99	n = 88	n = 110	
Lifestyle interventions	n = 153	n = 232	n = 194		n = 66	
RAAS inhibitors	n = 153	n = 116			n = 149	
Beta-blockers	n = 55	n = 110	n = 25			
EECP	n = 181			n = 131		
Long-acting nitrates	n = 55		n = 19			n = 4375
Neuromodulation	n = 58	n = 69	n = 57			
Ranolazine	n = 80	n = 109	n = 147		n = 47	
Trimetazidine	n = 129	n = 85				
Statins		n = 86				n = 7222
Traditional Chinese medicine	n = 271	n = 104				
Hormone therapy		n = 94				
Stem cell therapy	n = 40	n = 40	n = 40	n = 40		

**Table 4 jcm-14-04069-t004:** Assessment of included studies for primary outcome. RoB2 = risk of bias 2, D1 = randomisation process, S = bias arising from period and carryover effects, D2 = deviations from intended interventions, D3 = missing outcome data, D4 = measurement of the outcome, D5 = selection of the reported result, NOS = Newcastle–Ottawa Quality Assessment Scale. **Green** = low risk, **Orange** = some concerns, **Red** = high risk.

**RoB 2**							
Study	D1		D2	D3	D4	D5	Overall
Jansen et al. [2]							
Bove et al. [29]							
Sugisawa et al. [28]							
Asbury et al. [34]							
Michelsen et al. [35]							
Cao et al. [82]							
Oikawa et al. [16]							
Chen et al. [43]							
Villano et al. [74]							
Ma et al. [80]							
Asbury et al. [35]							
Boldueva et al. [75]							
Zhang et al. [91]							
**RoB 2—cross-over**							
Study	D1	S	D2	D3	D4	D5	Overall
Lanza et al. [18]							
Leonardo et al. [51]							
Prida et al. [22]							
Wu et al. [56]							
Gelman et al. [24]							
Lanza et al. [67]							
Shimizu et al. [52]							
Merz et al. [73]							
Kanadaşi et al. [44]							
Özçelik et al. [17]							
Cannon et al. [25]							
Nalbantgil et al. [78]							
**NOS**							
Study	Selection		Comparability		Outcomes		Overall
Sgueglia et al. [65]	3		1		3		Good
Jessurun et al. [66]	2		1		2		Fair
Mao et al. [83]	3		1		2		Good
Lanza et al. [68]	2		1		3		Good
Cunningham et al. [37]	2		1		3		Good
De Vries et al. [64]	2		1		2		Fair
Henry et al. [88]	2		1		3		Good
Noroozi et al. [79]	2		1		2		Fair
Galin et al. [76]	2		1		2		Fair
Corban et al. [89]	3		1		3		Good
Ashokprabhu et al. [90]	3		1		3		Good

## Data Availability

The data underlying this article are available in the article and in its online Appendix A.

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
