# Peer review of "Treatment Modalities for Angina with Non-Obstructive Coronary Arteries (ANOCA): A Systematic Review and Meta-Analysis"

_jcm, 2025, doi:10.3390/jcm14124069_

Round 1
Reviewer 1 Report
Comments and Suggestions for Authors
This systematic review and meta-analysis evaluates treatment efficacy for angina with ANOCA, a condition affecting 40% of angina patients undergoing CAG. By synthesizing data from 80 studies (10 aligned with current ANOCA definitions), the authors demonstrate significant improvements in angina frequency and any outcomes across therapies. Key contributions include a comprehensive overview of treatment effects and an urgent call for standardized ANOCA criteria in future trials. This study emphasizes addressing the heterogeneity of definition, while its limitation is the scarcity of high-quality, definition-consistent studies. This work highlights the need for robust RCTs to validate therapies for this understudied population.
- I think the author should provide the specific retrieval strategy of at least one database.
- "Pubmed" should be written as "PubMed" in the abstract and the main text.
- The first sentence of the introduction is not fluent enough, and the expression is ambiguous.
- This study is difficult to determine the impact of different treatment regimens (including different combination medications/treatment methods) on the outcome. The conclusion drawn by the authors is rather general, and this limitation needs to be emphasized.
I think the English language in this study is appropriate and understandable.
Reviewer 2 Report
Comments and Suggestions for Authors
This meta-analysis aimed to evaluate the efficacy of current treatment modalities for patients with ANOCA.
This meta-analysis showed a significant improvement in angina pectoris frequency in the majority of included treatment modalities, such as neuromodulation, trimetazidine, traditional Chinese medicine, beta-blockers, enhanced external counterpulsation, stem cell therapy, lifestyle interventions, RAAS-inhibitors and calcium channel blockers.
The strongest reduction in angina frequency was found in treatment with neuromodulation, while statins allowed to obtain an improvement in the exercise capacity. Overall, the biggest improvement in the quality of life was obtained with neuromodulation.
The mansucript is well written, the statistical analysis is adequate, the tables and figures are well done, the references are appropriate and the conclusions are supported by the results of the meta-analysis.
I have one only suggestion for the authors.
In the Limitations section, the authors could also discuss the potential causes of false positive exercise test in the included studies. It is important to consider that a large number of the included studies analyzed females with chest pain and positive exercise test, however the subsequent coronary angiography was normal. In this regard, the authors could discuss the sub-optimal specificity of exercise test for CAD detection (PMID: 2530605). They could mention and discuss important causes of false positive exercise test results (ST-segment changes simulating myocardial ischemia), such as exaggerated atrial repolarization waves (Ta wave), hyperventilation and mitral valve prolapse. In clinical practice, these individuals are commonly found with concave-shaped chest wall conformation and/or variuous degrees of pectus excavatum, symptomatic for chest pain and/or anxiety, with no evidence of obstructive CAD on CT scan or coronary angiography. Recent evidence indicates that a preliminary noninvasive chest shape assessment may allow the clinicians to identify, among the individuals with suspected CAD, those with a low pre-test probability (PTP) of obstructive CAD and good prognosis over a mid-to-long term follow-up period (PMID: 34485034). It is important to avoid unnecessary invasive examinations for individuals with low PTP of CAD, atypical chest pain, anxiety, commonly associated with anterior sternal depression and/or extrinsix thoracic compression on cardiac chambers.
